

# *Lanspora dorisauae*, a new marine fungus from rocky shores in Taiwan

Ka-Lai Pang[1], Sheng-Yu Guo[1], Ami Shaumi[1], Satinee Suetrong[2],
Anupong Klaysuban[2], Michael W. L. Chiang[3] and E. B. Gareth Jones[4]

[1] Institute of Marine Biology and Centre of Excellence for the Oceans, National Taiwan Ocean University, Keelung, Taiwan (ROC)
[2] Mycology Laboratory, Bioresources Technology Unit, National Center for Genetic Engineering and Biotechnology (BIOTEC), Khlong Luang, Pathumtani, Thailand
[3] Department of Chemistry, City University of Hong Kong, Hong Kong, SAR
[4] Department of Botany and Microbiology, College of Science, King Saud University, Riyadh, Kingdom of Saudi Arabia

Corresponding author
Ka-Lai Pang,
klpang@mail.ntou.edu.tw

## ABSTRACT

This article reports a new marine fungus, *Lanspora dorisauae* (*Phomatosporales*, Sordariomycetes, Ascomycota), on trapped wood collected in coastal sites of Taiwan. This new fungus was subjected to a morphological examination and a phylogenetic study based on a combined analysis of the 18S, 28S, ITS rDNA, TEF1-α and RPB2 genes. *Lanspora dorisauae* is characterized by dark-coloured ascomata with a short neck, periphysate ostioles, subclavate, deliquescing asci without an apical ring, presence of wide paraphyses, striated wall ascospores with crown-like appendages on one pole of the ascospores. Phylogenetically, *L. dorisauae* grouped with *Lanspora coronata* (type species) with strong support. *Lanspora coronata* lacks paraphyses and appendages occur on both ends of the ascospores, while paraphyses are present and ascospore appendage is unipolar in *L. dorisauae*. *Lanspora cylindrospora* formed a sister clade with *L. coronata* and *L. dorisauae*, but it significantly differs in morphology with the latter two species in having cylindrical asci with an apical J-ring, smooth ascospore wall and no ascospore appendages, and may be better referred to a new genus. *Lanspora*, together with *Phomatospora* and *Tenuimurus*, belong to the *Phomatosporaceae*, *Phomatosporales*. *Phomatospora berkeleyi* should be sequenced to test the validity of the order *Phomatosporales* and the family *Phomatosporaceae*.

## INTRODUCTION

The genus *Lanspora* was created to accommodate a marine fungus (*L. coronata*) found on driftwood collected from rocky beaches in Seychelles (*Hyde & Jones, 1986*). *Lanspora coronata* was originally placed in the *Halosphaeriaceae* (*Microascales*, Ascomycota) due to its perithecial ascomata, deliquescing asci and ascospores with crown-like appendages formed by fragmentation of the exosporium (*Hyde & Jones, 1986*). In a phylogenetic analysis based on five genes (18S, 28S ribosomal DNA (rDNA), elongation factor 1-alpha (EF1α), RNA polymerase II largest subunit (RPB1), RNA polymerase II second largest subunit (RPB2)), *L. coronata* grouped with *Ophiostoma piliferum* (*Ophiostomataceae*,

*Ophiostomatales*) (*Spatafora et al., 2006*), rather than in the *Halosphaeriaceae* as proposed by *Hyde & Jones (1986)*. *Lanspora coronata* differs in having cylindrical or oblong-ventricose asci and the presence of periphyses from taxa of the *Ophiostomatales* which are characterized by globose to ovoid asci and the absence of periphyses (*Alexopoulos, Mims & Blackwell, 1996*). Many species of the *Ophiostomatales* are insect-associated fungi.

In a combined 18S, 28S and ITS (internal transcribed spacers) rDNA phylogeny, *Lanspora* (*L. coronata*) formed a generally well-supported clade with *Phomatospora* and *Tenuimurus* (*T. clematidis*), and consequently, a new order *Phomatosporales* and a new family *Phomatosporaceae* were establish to accommodate these three genera (*Senanayake et al., 2016*). A new species *Lanspora cylindrospora* with a clypeus, presence of periphyses, cylindrical asci with a J- ring and cylindrical ascospores lacking appendages was described (*Hyde et al., 2020*), but these characteristics are not found in *L. coronata*. However, *L. cylindrospora* grouped with *L. coronata* in a phylogenetic tree based on sequences of the 18S, 28S, ITS rDNA, TEF1α (translation elongation factor 1-alpha) and RPB2 (*Hyde et al., 2020*).

Taiwan has a tropical/subtropical climate, but marine fungi of the temperate zone can be commonly found in northern Taiwan (*Pang & Jheng, 2012a*), where many new species of marine fungi were described, *e.g.*, *Pileomyces formosanus* (*Pang & Jheng, 2012b*) and *Sclerococcum vrijmoediae* (as *Dactylospora vrijmoediae*) (*Pang et al., 2014*). During our continued survey of lignicolous marine fungi in Taiwan, we discovered a marine fungus on driftwood/trapped wood in various rocky shores in Taiwan that is morphologically and phylogenetically (based on five genes) related to *L. coronata*, and is described here as new.

## MATERIALS AND METHODS

### Sample collection

Trapped wood was collected at a rocky shore (25°14′41.9″N 121°38′04.2″E) in Jin-Shan District, New Taipei City, Taiwan on 1 June 2022, placed on Ziplock plastic bag and transported to the laboratory at National Taiwan Ocean University, Keelung (*Pang et al., 2023*). The wood was cleaned briefly in running water, incubated on a tray and kept in a Ziplock plastic bag at room temperature (~25 °C) under a 12:12 light dark cycle. The wood was checked periodically for the growth of marine fungi.

### Morphological characterization

The methods used in *Pang et al. (2013)* was followed for the morphological examination of the new fungus. For sectioning of ascomata, wood blocks with ascomata were cut out and fixed in FAA solution (5% formaldehyde and 5% glacial acetic acid in 50% ethanol) overnight at 4 °C. The wood blocks were washed three times in 50% ethanol dehydrated in a graduated t-butanol/ethanol/water series (10/40/50, 20/50/30, 35/50/15, 55/45/0, 75/25/0, 100/0/0, 100/0/0, in percentage) and embedded in paraffin. Paraffin sections were cut on a ERM_200P rotary microtome (ERMA, Saitama, Japan), floated on 42 °C water-bath and mounted on microscope slides. Dried sections were deparaffinised and rehydrated through a graded series of ethanol. The sections were stained with 1% safranin O in 50% ethanol

(10 s) and 0.5% Orange G in 95% ethanol (30 s). After washing and dehydration, permanent slides were made for the stained sections in Permount (Fisher, Waltham, MA, USA).

For examination of asci and ascospores, ascomata were observed using an Olympus SZ61 stereomicroscope (Tokyo, Japan), cut opened with a razor blade, picked up with a fine forceps, and mounted on a slide in sterile seawater.

Structures of ascomata, asci and ascospores were examined using an Olympus BX51 microscope (Tokyo, Japan), with photographs taken with an Olympus DP20 Microscope Camera (Tokyo, Japan).

## Single spore isolation

A spore suspension was prepared by transferring a spore mass of the new fungus to a drop of sterile natural seawater on a sterilized glass slide with a flame-sterilized forceps with mixing. This spore suspension was dropped onto the surface of cornmeal seawater agar (CMAS; 17 g cornmeal agar (Difco), 1 L natural seawater, 0.5 g/L each of Penicillin G sodium salt and streptomycin sulfate) plate. The plate was incubated at 25 °C until the spores were germinated. Single germinated spores were picked up, transferred onto fresh CMAS plates, and incubated at 25 °C. The ex-type culture was kept at Bioresource Collection and Research Center, Hsinchu, Taiwan.

## Molecular analysis

Aerial mycelia of the new fungus were scraped off from the CMAS plate with a flame-sterilized spatula, and ground into powder in a mortar and pestle. Total DNA was extracted using the DNeasy Plant Mini Kit (Qiagen, Valencia, CA, USA) according to the manufacturer's instructions. Extracted DNA (1 µL) was used directly for PCR reactions with the following ingredients: 0.2 µM of each primer (18S rDNA: NS1/NS2, ITS rDNA: ITS5/ITS4 (*White et al., 1990*), 28S rDNA: LROR/LR6 (*Vilgalys & Hester, 1990*; *Bunyard, Nicholson & Royse, 1994*), TEF1α: EF1-983F/EF1-2218R (*Rehner & Buckley, 2005*), RPB2: fRPB2-5F/fRPB2-7cR (*Liu, Whelen & Hall, 1999*)), 0.5 volume of Gran Turismo PreMix (Ten Giga BioTech, Taiwan) and topped up to 25 µL with PCR water. The amplification cycle consisted of an initial denaturation step of 94 °C for 5 min followed by 35 cycles of (i) denaturation (94 °C for 0.5 min), (ii) annealing (55 °C for 0.5 min) and (iii) elongation (72 °C for 0.5 min) and a final 11 min elongation step at 72 °C. The PCR products were shipped to Tri-I Biotech, Inc., Taiwan, for purification and Sanger sequencing with the same primers (both strands) described above.

Returned sequences were checked for ambiguity, assembled and deposited in GenBank (accession nos. in Table 1). An initial nucleotide BLAST search suggested that the sequences were affiliated with taxa of the *Phomatosporales*. These sequences were aligned with the 18S, 28S, ITS, TEF1a and RPB2 genes of the taxa in the *Phomatosporales* and *Ophiostoma piliferum* and *O. stenoceras* (outgroup taxa) in the program MUSCLE (*Edgar, 2004*) in MEGA11 (*Tamura, Stecher & Kumar, 2021*). The final dataset contained 16 sequences and a total of 7,623 nucleotide positions. The genes were analyzed simultaneously. A maximum likelihood analysis including bootstrapping was performed

in MEGA11 (*Tamura, Stecher & Kumar, 2021*) with the following settings: 500 bootstrap, General Time Reversible model (GTR), gamma distributed (G), number of discrete gamma categories set at 5, heuristic search with Nearest-Neigbor-Interchange, initial tree from NJ/BioNJ method, branch swapping strong. A maximum parsimony (MP) analysis including bootstrapping was also run in MEGA11 with the following settings: 500 bootstrap, Tree-Bisection-Reconnection (TBR), number of initial trees (random addition) = 5, MP search level = 1, maximum number of trees to retain = 100.

For Bayesian analysis, BEAUti v1.10.4 was used for prior settings and the dataset was analyzed in BEAST v1.10.4 (*Suchard et al., 2018*). The following analytical settings were used for the analysis: GTR, estimated base frequency, gamma distribution, number of gamma categories set at five, a strict clock, Coalescent: Constant Size as the speciation model, running 10 million generations with parameters and trees sampled every 1,000 generations. The first 10% of the trees were discarded as the burn-in based on the effective sample size (ESS) of the parameter statistics in Tracer v1.7.2 (*Rambaut et al., 2018*). A summary tree was generated in TreeAnnotator v1.10.4 (*Suchard et al., 2018*) and viewed and edited in FigTree v1.4.4 (available at https://github.com/rambaut/figtree/releases) (Supplemental file).

The electronic version of this article in Portable Document Format (PDF) in a work with an ISSN or ISBN will represent a published work according to the International Code of Nomenclature for algae, fungi, and plants, and hence the new names contained in the electronic publication of a PeerJ article are effectively published under that Code from the electronic edition alone, so there is no longer any need to provide printed copies.

In addition, new names contained in this work have been submitted to MycoBank from where they will be made available to the Global Names Index. The unique MycoBank number can be resolved and the associated information viewed through any standard web browser by appending the MycoBank number contained in this publication (MB 847455) to the prefix http://www.mycobank.org/MB/. The online version of this work is archived and available from the following digital repositories: PeerJ, PubMed Central and CLOCKSS.

# RESULTS

Based on the phylogenetic analysis of five genes (18S, 28S, ITS rDNA, TEF1α and RPB2), the new species *L. dorisauae* formed a strongly supported clade with *L. coronata* while *L. cylindrospora* formed a sister relationship with these two species (Fig. 1). *Lanspora dorisauae* is similar to *L. coronata* in having dark-coloured, coriaceous ascomata, subclavate asci, elongate-ellipsoidal ascospores with striated wall and crown-like ascospore appendages (*Hyde & Jones, 1986*). However, *L. dorisauae* differs from *L. coronata* in the lack of periphyses and the presence of crown-like appendages at one end of the ascospores in the former species. Also, paraphyses are present in *L. dorisauae*. *Lanspora dorisauae* was collected at several coastal locations in Taiwan, suggesting that it is a common marine fungus.

**Table 1 GenBank accession numbers.** Sequences used in the phylogenetic analysis in this study and their GenBank accession numbers.

| Species | Culture number | GenBank accession number | | | | |
|---|---|---|---|---|---|---|
| | | 18S rDNA | 28S rDNA | ITS rDNA | EF1α | RPB2 |
| *Lanspora dorisauae* sp. nov. | NTOU3803 | OR046876 | OR046883 | OR046882 | OR141138 | — |
| *Lanspora dorisauae* sp. nov. | BCRC FU30316 | OQ130044 | OQ130043 | OQ130045 | OQ570968 | OQ570969 |
| *Lanspora coronata* | AFTOL-ID 736 | DQ470996 | — | — | DQ471067 | DQ470899 |
| *Lanspora coronata* | JK4839A | U48424 | U46889 | — | — | — |
| *Lanspora cylindrospora* | NFCCI4391 | — | — | — | MN795088 | MN795090 |
| *Lanspora cylindrospora* | NFCCI4427 | — | MN168892 | MN168890 | MN795089 | — |
| *Lanspora cylindrospora* | NFCCI4665 | MN169053 | MN168891 | MN168889 | — | — |
| *Ophiostoma piliferum* | AFTOL-ID 910 | DQ471003 | DQ470955 | — | DQ471074 | DQ470905 |
| *Ophiostoma stenoceras* | AFTOL-ID 1038 | DQ836897 | DQ836904 | — | FJ190618 | DQ836891 |
| *Phomatospora bellaminuta* | AFTOL-ID 766 | FJ176803 | FJ176857 | — | — | FJ238345 |
| *Phomatospora biseriata* | MFLUCC 14-0832A | KX549458 | KX549448 | KX549453 | — | — |
| *Phomatospora biseriata* | MFLUCC 14-0832B | KX549459 | KX549449 | KX549454 | — | — |
| *Phomatospora striatigera* | CBS 133932 | — | KM213618 | KM213617 | — | — |
| *Phomatospora viticola* | MFLU16-1973 | — | KX549452 | KX549457 | — | — |
| *Tenuimurus clematidis* | MFLU16-1971 | — | KX549451 | KX549456 | — | — |
| *Tenuimurus clematidis* | MFLUCC14-0833 | — | KX549450 | KX549455 | — | — |

## Taxonomy

*Lanspora dorisauae* K. L. Pang, Suetrong, M. W. L. Chiang and E. B. G. Jones, sp. nov.
Fig. 2
**Mycobank (MB#847455).**
**GenBank (LSU = OQ130043, ITS = OQ130045, SSU = OQ130044, TEF1α = OQ570968, RPB2 = OQ570969).**
Saprobic on trapped wood on a rocky shore. **Sexual morph** Ascomata 148–213 μm high, 240–347 μm diam. ($\bar{X}$ = 174 × 277 μm, *n* = 4), immersed, subglobose to ellipsoidal in side-view, solitary to gregarious, coriaceous, brown to black, papillate (Fig. 2A). Necks short, 44–55 μm long, 80–109 μm diam. ($\bar{X}$ = 48 × 99 μm, *n* = 4), cylindrical, brown to black (Fig. 2A). Periphyses not observed. Peridium 13–24 μm ($\bar{X}$ = 19 μm, *n* = 4), equal in thickness, comprising one stratum of multilayers (over five layers) of brown cells of *textura angularis* with large lumina (Fig. 2B). Paraphyses 74 × 9 μm (Fig. 2C). Asci 41–51 × 16 μm ($\bar{X}$ = 44 × 16 μm, *n* = 4), 8-spored, unitunicate, thin-walled, clavate, pedunculated (Fig. 2D). Ascospores 8–13 × 5–7 μm ($\bar{X}$ = 11 × 6 μm, *n* = 35), biseriate, overlapping, unicellular, broadly ellipsoidal with one half broader than the other, with longitudinal wall striations, guttulate, appendaged (Figs. 2E and 2F). Appendages at one end of ascospores, five to seven in number, crown-like, sheet-like irregular and radiating, delicate and subgelatinous (Fig. 2G). **Asexual morph** Undetermined.
**Culture characteristics**: Colonies attained 1 cm in diameter after 1 week of incubation at 25 °C in darkness, white to pale yellow; mycelia sparse, feathery, with no pigment formed in the agar.
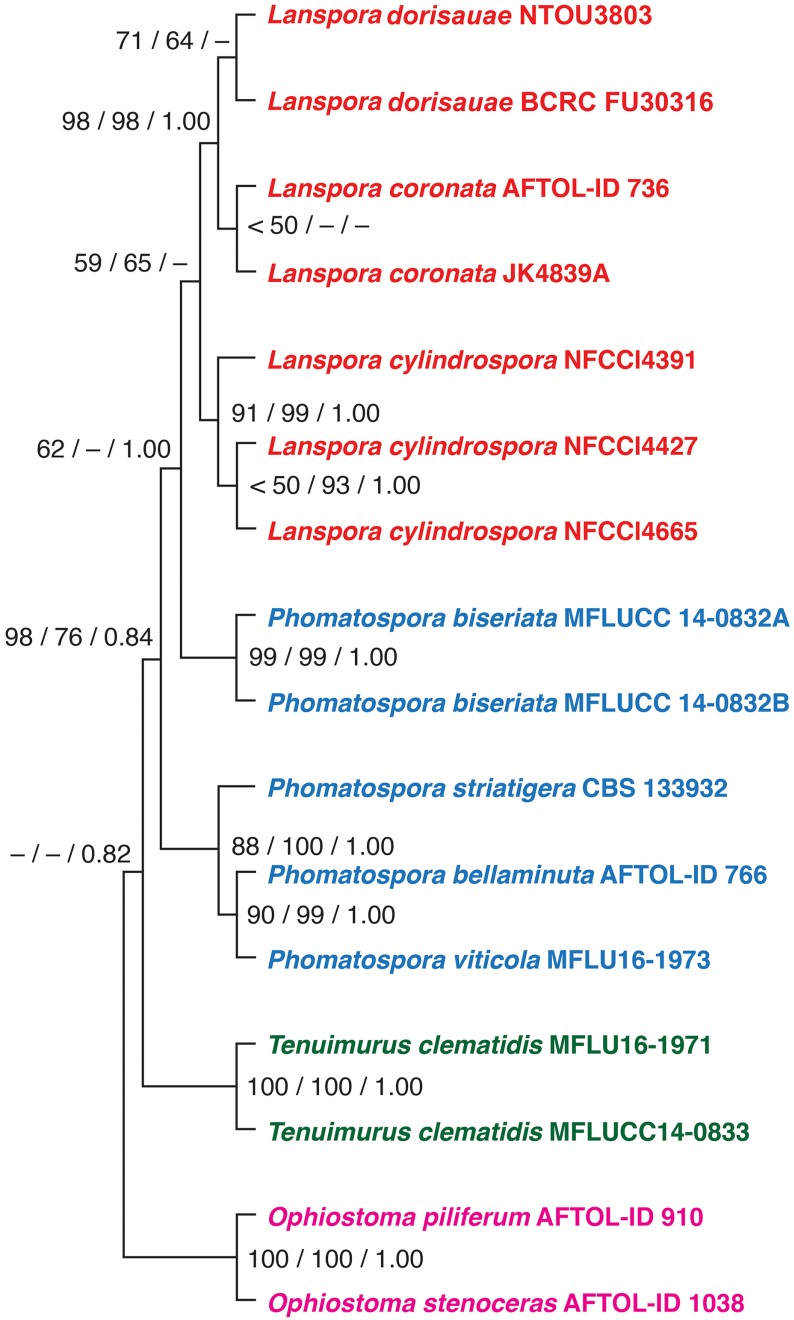

**Figure 1 The phylogenetic tree.** The single most parsimonious tree produced from maximum parsimony analysis of a combined dataset of five genes (18S, 28S, ITS rDNA, EF1a, RPB2). The numbers at the nodes represent maximum parsimony bootstrap, maximum likelihood bootstrap and posterior probability, respectively.                           

**Holotype**: TAIWAN: Jin-Shan. On a piece of unidentified trapped wood, 1 June 2022, S. Y. Guo and K. L. Pang, F26641 (National Museum of Natural Science Herbarium, Taichung, Taiwan), dried wood.

**Ex-type culture:** BCRC FU30316 (Bioresource Collection and Research Center, Hsinchu, Taiwan).

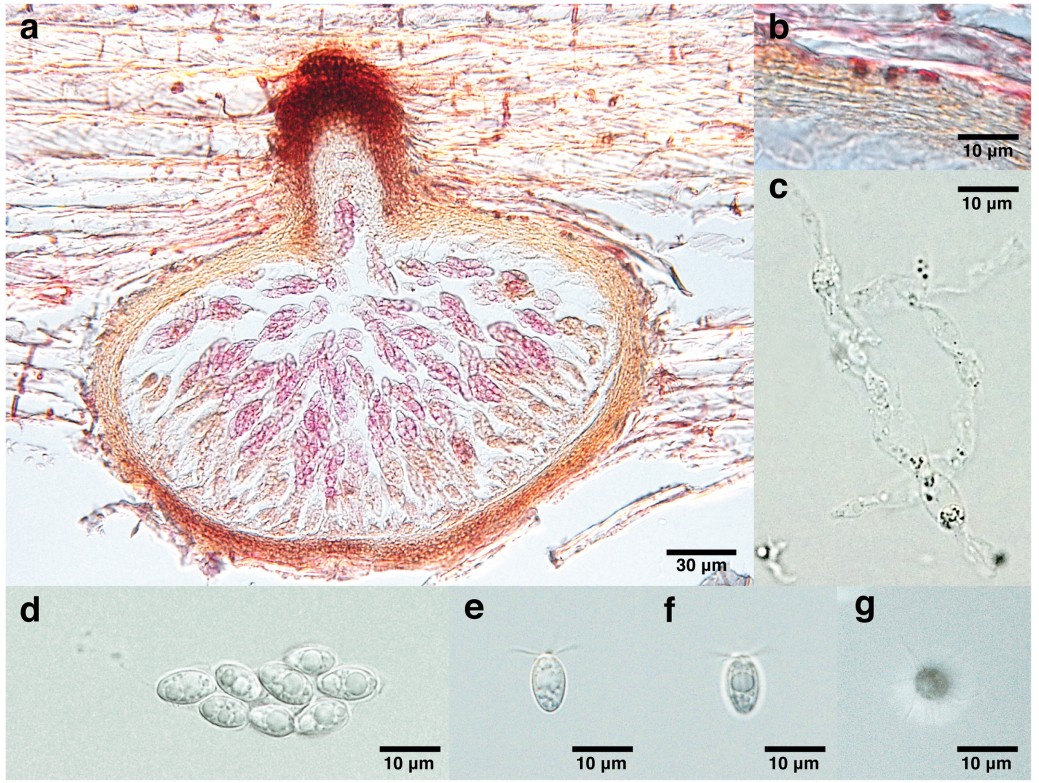

**Figure 2** *Lanspora dorisauae.* (A) Section of ascoma. (B) Peridium made of rows of cells of *textura angularis* with large lumina. (C) Paraphyses. (D) Clavate ascus. (E) Broadly ellipsoidal ascospores with oil guttules. (F) Longitudinal striations on ascospore wall. (G) Radiating appendages at one end of ascospore.

**Other culture:** NTOU3803 (National Taiwan Ocean University Culture Collection).
**Etymology**: In memory of Dr. Doris Wai-Ting Au, a marine zoologist/mycologist who taught me (K. L. Pang) the right way to do scientific research.
**Distribution**: Jin-Shan, Li-lao, Ying-Ke-Shih (New Taipei City), Shih-Ti-Ping (Hualien County) (Taiwan).

## DISCUSSION

Currently, three genera are included in the *Phomatosporaceae*, *Phomatosporales*: *Lanspora*, *Tenuimurus* and *Phomatospora*. Morphologically, *Phomatospora* (type species: *P. berkeleyi*) and *Tenuimurus* (type species: *T. clematidis*) are similar in having cylindrical, persistent asci with an apical J- ring, smooth-walled, ellipsoidal ascospores (*Fallah & Shearer, 1998*; *Senanayake et al., 2016*). *Lanspora* (*L. coronata* and *L. dorisauae*) differs from these two genera in having subclavate, deliquescing asci and striated ascospores with crown-like appendages (*Hyde & Jones, 1986*). Although *L. cylindrospora* phylogenetically is related to *L. coronata*, the former species morphologically differs significantly from the latter in having cylindrical asci with a J- apical ring and ascospores without appendages (*Hyde et al., 2020*). In contrast, *L. coronata* and *L. dorisauae* both have clavate, deliquescing asci without an apical ring and ascospores with crown-like appendages (*Hyde*

& Jones, 1986; this study). *Lanspora coronata* lacks paraphyses and appendages occur on both ends of the ascospores, while paraphyses are present and ascospore appendage is unipolar in *L. dorisauae*. *Lanspora cylindrospora* may be better referred to a new genus based on its significant morphological differences with *L. coronata* and *L. dorisauae*. Additional genes should be sequenced for the isolates of *L. cylindrospora* to provide a robust phylogeny to establish this taxonomic change.

No sequences of *P. berkeleyi* were available in the GenBank for the phylogenetic analysis when the new family *Phomatosporaceae* (the new order *Phomatosporales*) was established to include *Lanspora*, *Tenuimurus* and *Phomatospora* (Senanayake et al., 2016). *Phomatospora berkeleyi* should be isolated and sequenced to evaluate the validity of *Phomatosporales* and *Phomatosporaceae*.

In conclusion, the new marine fungus *L. dorisauae* is morphologically similar to *L. coronata*, the type species of the genus *Lanspora*. The phylogenetic analysis based on a combined analysis of 18S, 28S, ITS rDNA, TEF1α and RPB2 genes confirmed the placement of *L. dorisauae* in *Lanspora* (*Phomatosporaceae*, *Phomatosporales*). *Lanspora dorisauae* grows on wood and fulfill a decomposer role in the marine environment.

### Funding
The authors received no funding for this work.

### Competing Interests
The authors declare that they have no competing interests.

### Author Contributions

- Ka-Lai Pang conceived and designed the experiments, performed the experiments, analyzed the data, prepared figures and/or tables, authored or reviewed drafts of the article, and approved the final draft.
- Sheng-Yu Guo conceived and designed the experiments, performed the experiments, analyzed the data, prepared figures and/or tables, authored or reviewed drafts of the article, and approved the final draft.
- Ami Shaumi conceived and designed the experiments, analyzed the data, prepared figures and/or tables, authored or reviewed drafts of the article, and approved the final draft.
- Satinee Suetrong conceived and designed the experiments, analyzed the data, authored or reviewed drafts of the article, and approved the final draft.
- Anupong Klaysuban conceived and designed the experiments, analyzed the data, authored or reviewed drafts of the article, and approved the final draft.
- Michael W. L. Chiang conceived and designed the experiments, performed the experiments, analyzed the data, prepared figures and/or tables, authored or reviewed drafts of the article, and approved the final draft.

- E. B. Gareth Jones conceived and designed the experiments, analyzed the data, authored or reviewed drafts of the article, and approved the final draft.

## DNA Deposition

The following information was supplied regarding the deposition of DNA sequences:

The specimen is available at Mycobank: MB #847455.

The data are available at GenBank: LSU, OQ130043; ITS, OQ130045; SSU, OQ130044; TEF1α, OQ570968; RPB2, OQ570969.

## Data Availability

The raw data are available in the Supplemental Files.

## New Species Registration

The following information was supplied regarding the registration of a newly described species:

*Lanspora dorisauae* (*Phomatosporales*, Sordariomycetes, Ascomycota): MycoBank #847455.

## Supplemental Information

Supplemental information for this article can be found online at http://dx.doi.org/10.7717/peerj.15958#supplemental-information.

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
