# Peer review of "Lanspora dorisauae, a new marine fungus from rocky shores in Taiwan"

_PeerJ, doi:10.7717/peerj.15958_

## Round 0.1 · original submission · Major Revisions

Two experts assessed your manuscript and found merit for publication in this journal. Both reviewers agreed that the experimental design is adequate but several details are missed. Please address all concerns in a revised version of the manuscript.

·

Basic reporting

Although the authors describe a well-justified new species of the genus Lanspora, the quality of the text is poor and needs to be carefully revised, even in some basic aspects regarding taxonomy, which are commented on in the pdf file attached.

Experimental design

The experimental design is correct, but details on the methods described are missing in some procedures (see the annotated file).

Validity of the findings

Part of the abstract must be rewritten according to the journal criterium (see comments in the same pdf).

Introduction and Discussion (first part) are too repetitive.

The quality of the tree is very poor.

Accession numbers of the new fungus are difficult to follow regarding the material submitted (ms, table, phylotree).

Additional comments

Specific comments
Lin 43. Expand names of the gene markers
Lin 63. Check title
Lin 69. Introduce incubation conditions
Lin 71. Was the fungus checked for growth in vitro? Please include some information on this subject.
Lins 84-87. Rewrite this part of the paragraph.
Lins 112-114. It is highly recommended to deposit alignments.
Lins 146-155. “Culture characteristics” The data included here does not correspond to that. Please verify comparing with other similar publications.
Lins 158-168. The first paragraph in Discusion is very repetitive regarding Introduction. So, it must be abbreviated o deleted.

Figures.
The format and style of the tree it is really very poor. I encourage authors to replace it with a more attractive one that distinguishes the new species from the rest of the species included in the analysis.
The type strains should be indicated for all the species if they are used for the analysis.

Table 1.
It is not necessary to repeat the names of the species. Check!
The type strains for all the species should be indicated in the table.

Relevant!!
No collection number included in the tree neither in the table corresponding to the new species matches with the collection number assigned to the ex-type strain, which is mentioned in the text (see line 150). Please unify, otherwise the information of the isolates is very confusing for the readers.

·

Basic reporting

The English language is good, however, the writing needs to be improved in several paragraphs of the text (this is pointed out in the attached file). The bibliographic references are related to the research topic, but I consider that for an article describing a new species, they are too few. A more exhaustive search on the subject should be made. The article has an orderly structure and in each section the most relevant points are mentioned. The legends of the tables and figures should be improved.

Experimental design

It is an original and novel research, however, from the beginning, the objective of the work and the research question are not defined. These comments were put in more detail in the attached file. The description of the sample collection is clear and concise. However, it would be useful to provide more details on the exact geographic location of the rocky shores where the trapped wood pieces were collected. This would help to better contextualize the study and to understand the provenance of the samples. The morphological characterization of the fruiting bodies is adequately described, mentioning the microscopes and camera used to observe and photograph the structures. This provides information on the methods used for visual analysis of fungal characteristics. As for molecular analysis, the steps for DNA extraction and amplification of specific genes are mentioned. However, the wording could be clearer and more precise in this section. It is mentioned that PCR products were sent for sequencing, but it would be useful to provide additional information on the sequencing method used and any quality criteria applied to the resulting sequences. As for the phylogenetic analysis, the general steps and tools used are described. However, the wording could be clearer in describing the specific settings and parameters used in the maximum likelihood, parsimony and Bayesian analyses. It would be useful to provide more details on the options selected for each type of analysis and how the results were interpreted. In addition, it would be important to mention the tools or programs used for assembling and editing the sequences and constructing the phylogenetic tree.

Validity of the findings

In terms of contribution to scientific knowledge, the description of a new species and taxonomic discussion provide valuable information on fungal diversity in marine environments. This may help to broaden our understanding of the ecology and evolution of fungi in these specific habitats. Furthermore, by discussing the phylogenetic relationship of Lanspora to other taxa and proposing taxonomic changes, the work may influence future research and classification of these organisms. Although studies on marine fungi exist, knowledge about fungal diversity in marine habitats is still developing, and every contribution that expands our understanding in this field is valuable.
Phylogenetic analyses based on five genes (18S, 28S, ITS rDNA, EF1-α and RPB2) were performed to investigate the evolutionary relationships between Lanspora dorisauae, L. coronata and L. cylindrospora. This phylogenetic approach provides information on the genetic relationship between these species and may help to understand their evolution and classification.
Regarding the robustness of the data presented, it is important to consider the methodology used and the amount of evidence supporting the results. According to the information provided, phylogenetic analyses were performed based on five genes, which is a commonly accepted and robust approach in studies of this type. However, we do not have specific details on the sequencing and analysis methods used in this particular study. These should be better explained.
With respect to the conclusions this article does not present.

Additional comments

Overall, the title is informative and captures the reader's attention by mentioning the novelty of the discovery and the specific geographic location. The abstract clearly and concisely presents the key aspects of the discovery of the marine fungus Lanspora dorisauae, including its morphological characteristics, phylogeny and possible taxonomic implications.
In the case of the introduction, the language used seems clear and concise for the most part. However, there are some sentences that could be rewritten to improve fluency and clarity (in the attached file you will find what they are). The information provided in the introduction is adequate to present the context of the study on the new marine fungus. The genus Lanspora and its initial discovery in Seychelles is mentioned, as well as previous taxonomic classifications and phylogenetic analyses that have taken place. However, it is necessary to provide more information, add a brief explanation or definition of the technical terms used, so that readers unfamiliar with scientific terminology can better understand the characteristics of the fungus. It is mentioned that a new marine fungus has been discovered on the rocky shores of Taiwan, which is morphologically similar to L. coronata and phylogenetically related to it. However, I feel it is necessary to provide additional details on the morphological similarities and differences between the new fungus and L. coronata. It would be useful to add a sentence summarizing the main objective of the study or what is sought to be achieved by describing this particular new marine fungus.
The results presented include the key findings of the study, such as the phylogenetic relationship of the new species L. dorisauae with L. coronata and L. cylindrospora, as well as the distinctive morphological characteristics of L. dorisauae compared to L. coronata. The inclusion of morphological data, such as size and shape of ascomata, characters of microscopic structures, and characteristics of spores, helps to provide a detailed description of the new species and to distinguish it from related species. In terms of additional information that could be considered, it might be useful to include more details on the characteristics of the asexual morph, if they were observed in the study. Also, if there is any discussion of the ecology or ecological role of the species in its marine habitat, this could be added to provide a more complete picture. I consider that the discussion is poor and there is a lack of information to support that this species is new and does not correspond to a species already described.
It might be useful to discuss the ecological importance of these species in the marine habitat and how their discovery and characterization contribute to the knowledge of fungal diversity in that environment. Mention could also be made of any implications or relevance the results have in terms of biotechnology, medicine, or other related fields.

---

## Round 0.2 · accepted · Accept

The two Reviewers that originally revised your manuscript think that this version is now suitable for publication. I agree with this point of view, congratulations.

·

Basic reporting

The article has been improved considerably

References are correct

The article is well structured

A new species is proposed based on morphological and phylogentic data

Experimental design

They are fine

Validity of the findings

It is a taxonomic contribution of the genus Lansposa

Conclusions are well stated

Additional comments

In table 1, please avoid to repite the name of the species

·

Basic reporting

The text is well written. Comments about the language were addressed. The corrections made to the figures and tables were also taken into account.

Experimental design

The objective was incorporated in the introduction. In addition, the experimental strategy used is explained in more detail.

Validity of the findings

The results presented are related to the methodology used to fulfill the research question. In addition, it is a novel work, because a new species is being described. The conclusion is complete and justifies the findings found.

Additional comments

I believe that the authors took into account all the comments that were made about their research article.